# Sudachitin and Nobiletin Stimulate Lipolysis via Activation of the cAMP/PKA/HSL Pathway in 3T3-L1 Adipocytes

**DOI:** 10.3390/foods12101947

**Published:** 2023-05-10

**Authors:** Tomoki Abe, Tomoyuki Sato, Kazutoshi Murotomi

**Affiliations:** 1Healthy Food Science Research Group, Cellular and Molecular Biotechnology Research Institute, National Institute of Advanced Industrial Science and Technology (AIST), Ibaraki 305-8566, Japan; satou.tom@aist.go.jp; 2Molecular Neurophysiology Research Group, Biomedical Research Institute, National Institute of Advanced Industrial Science and Technology (AIST), Ibaraki 305-8566, Japan; k-murotomi@aist.go.jp

**Keywords:** sudachitin, nobiletin, adipocyte, lipolysis, hormone-sensitive lipase

## Abstract

Polymethoxyflavones are flavonoids that are abundant in citrus fruit peels and have beneficial effects on human health. Previous studies have demonstrated that the polymethoxyflavones, namely sudachitin and nobiletin, ameliorate obesity and diabetes in humans and rodents. Although nobiletin induces lipolysis in adipocytes, lipolytic pathway activation by sudachitin has not been clarified in adipocytes. In this study, the effect of sudachitin on lipolysis was elucidated in murine 3T3-L1 adipocytes. Glycerol release into the medium and activation of the cyclic AMP (cAMP)/protein kinase A (PKA)/hormone-sensitive lipase (HSL) pathway was evaluated in 3T3-L1-differentiated adipocytes. Treatment with sudachitin and nobiletin for 24 and 48 h did not induce cytotoxicity at concentrations of up to 50 μM. Sudachitin and nobiletin at concentrations of 30 and 50 μM increased intracellular cAMP and medium glycerol levels in 3T3-L1 adipocytes. Western blotting revealed that sudachitin and nobiletin dose-dependently increased protein levels of phosphorylated PKA substrates and phosphorylated HSL. Sudachitin- and nobiletin-induced glycerol release, phosphorylation of PKA substrates, and HSL phosphorylation were suppressed by pharmacological inhibition of adenylate cyclase and PKA. These findings indicated that sudachitin, similar to nobiletin, exerts anti-obesogenic effects, at least in part through the induction of lipolysis in adipocytes.

## 1. Introduction

Overweight and obesity are global public health challenges that have increased worldwide [1]. White adipose tissue (WAT) is the primary tissue responsible for lipid storage. Increased consumption of Western-style diets results in overnutrition, thereby leading to excess lipid accumulation in WAT [2]. Hypertrophic adipocytes accumulate in obese adipose tissue and cause systemic metabolic dysfunction mediated by insulin resistance and inflammatory adipokine secretion [3].

Citrus fruit contains various functional polyphenols, flavonoids, and alkaloids [4]. Citrus flavonoids are the main bioactive components that improve metabolic dysfunction in rodents and humans [5]. Citrus peel is a rich source of polymethoxyflavones, particularly nobiletin (3′,4′,5,6,7,8-hexamethoxyflavone) [6]. Several studies have revealed that consumption of nobiletin prevents diabetes, obesity, atherosclerosis, fatty liver, and non-alcoholic steatohepatitis in rodents [7,8,9,10,11]. Because nobiletin contributes to the regulation of various adverse events in metabolic syndrome, such as inflammation, oxidative stress, and lipid accumulation [12,13,14,15,16,17], nobiletin intake is hypothesized to have beneficial effects in humans. Sudachitin (5,7,4′-trihydroxy-6,8,3′-trimethoxyflavone), which is contained in the peel of *Citrus sudachi*, is also a polymethoxyflavone with hydroxyl groups at the C-5, C-7, and C-4′ positions, which are methoxy groups in nobiletin [18]. Consumption of sudachi peel extraction prevents excessive fat deposition. Treatment with sudachi peel extraction decreased intracellular lipid levels, mediated by AMP-activated protein kinase activation in murine C2C12 myotubes [19]. Consumption of sudachi peel extraction suppressed body weight gain and WAT inflammation in mice that were fed a high-fat diet (HFD) [20]. Sudachitin exhibits anti-inflammatory effects on various cell types, such as macrophages [21], osteoclast precursors [22], and periodontal ligament cells [23,24]. Recently, Shikishima et al., reported that sudachitin-containing sudachi peel extracts decrease the ratio of abdominal fat to subcutaneous fat and modestly decrease abdominal circumference [25]. Sudachitin improved hyperlipidemia and hyperglycemia in mice that were fed a HFD [26]. It is assumed that the improvement in metabolism is due to the increase in energy expenditure, as sudachitin increases mitochondrial biogenesis in murine skeletal muscles [27]. These findings suggest that sudachitin may contribute to reductions in overweight and obesity. However, the effect of sudachitin on lipid accumulation in adipocytes, a key event in obesity, remains unclear.

Lipid accumulation depends on the balance between lipolysis, lipogenesis, and lipid uptake [28]. Lipolysis, a process of lipid catabolism, is highly regulated by lipases (for example, adipose triglyceride lipase (ATGL) and hormone-sensitive lipase (HSL)) [29]. Lipolysis is inhibited by insulin and activated by catecholamine secretion induced by exercise and starvation. The main intracellular signaling pathway of lipolysis is the β3-adrenergic receptor (β3-AR)-mediated stimulation of adenylate cyclase (AC) and the subsequent induction of protein kinase A (PKA) phosphorylation, mediated via increased intracellular cyclic AMP (cAMP) levels in adipocytes [30]. PKA activation induces phosphorylation of Ser563 [31] and Ser660 in HSL [32]. Ser660 is the major activity-controlling site in HSL [33], and HSL phosphorylation at Ser660 is a critical response to the induction of lipolysis. Because natural nutrients, such as ferulic acid [34], lutein [35], and lipoic acid [36], induce HSL phosphorylation in adipocytes, we hypothesized that sudachitin is involved in lipolysis, mediated by HSL activation.

This study aimed to determine the effects of sudachitin on lipolysis in 3T3-L1 adipocytes, which are often used as a model to investigate lipid accumulation mechanisms. The findings indicated that sudachitin exerts its anti-obesogenic effects, at least in part, by inducing adipocyte lipolysis.

## 2. Materials and Methods

### 2.1. Chemicals and Reagents

Dexamethasone, fetal bovine serum (FBS), high-glucose Dulbecco’s Modified Eagle’s Medium (DMEM), insulin solution, 3-isobutyl-1-methylxanthine, oil red O solution, and PhosSTOP were purchased from Sigma-Aldrich Corporation (St. Louis, MO, USA). Dimethyl sulfoxide (DMSO), ImmunoStar LD, lactate dehydrogenase (LDH)-Cytotoxic Test Wako, 10% neutral-buffered formalin, nobiletin, and sudachitin were purchased from FUJIFILM Wako Pure Chemical Corporation (Tokyo, Japan). The structures of sudachitin and nobiletin are shown in Figure 1A. Cyclic AMP ELISA Kit, H-89, isoproterenol, and SQ22536 were purchased from Cayman Chemical (Ann Arbor, MI, USA). cOmplete Protease Inhibitor Cocktail was purchased from Roche Diagnostics (Indianapolis, IN, USA). Block Ace was purchased from DS Pharma Biomedical Co., Ltd., (Osaka, Japan). EnzyChrom^TM^ Glycerol Assay Kit was purchased from BioAssay Systems (Hayward, CA, USA).

### 2.2. Cell Culture and Differentiation of 3T3-L1 Cells

Murine 3T3-L1 cells (CL-173, ATCC, Manassas, VA, USA) were maintained and proliferated at 37 °C with 5% CO_2_ in DMEM supplemented with 10% FBS and 1% penicillin-streptomycin mixed solution. The medium was replaced every 2 days. At day 0 (2 days after 100% confluence), differentiation of the 3T3-L1 cells was induced in DMEM containing 10% FBS, antibiotics, 1 μM dexamethasone, 500 μM 3-isobutyl-1-methylxanthine, and 340 nM insulin. Two days later, the medium was replaced with DMEM supplemented with 10% FBS, antibiotics, and 340 nM insulin. At day 8, fully differentiated 3T3-L1 adipocytes were treated with sudachitin or nobiletin (FUJIFILM Wako Pure Chemical Corporation, Tokyo, Japan). 3T3-L1 adipocytes were pretreated with the following chemical inhibitors for 1 h prior to sudachitin and nobiletin treatment: 100 μM SQ22536 and 20 μM H-89 to inhibit AC and PKA, respectively.

### 2.3. Oil Red O Staining

Oil red O staining for the quantification of intracellular lipids was performed as described previously, with some modifications [37]. At day 8, mature 3T3-L1 adipocytes were treated with 50 μM sudachitin and nobiletin. After 48 h, adipocytes were washed thrice with phosphate-buffered saline (PBS) and fixed with 10% neutral-buffered formalin at 25 °C for 20 min. After washing thrice with PBS, the cells were incubated with oil red O solution at 25 °C for 20 min. After washing thrice with PBS, representative images of the stained adipocytes were captured using a BioZero BZ-X810 microscope (Keyence, Osaka, Japan). For quantification, the absorbance of extracted oil red O was detected using a microplate reader (iMark Microplate Reader, BioRad, Hercules, CA, USA) at 490 nm.

### 2.4. LDH Activity Assay

LDH release to the culture medium was measured to assess cell damage [38]. 3T3-L1 adipocytes on day 8 were treated with different sudachitin and nobiletin concentrations (1–50 μM). Adipocytes treated with 0.1% DMSO served as vehicle controls. After 24 and 48 h, the media were collected from each well, and LDH activity was measured using the LDH-Cytotoxic Test Wako (FUJIFILM Wako Pure Chemical Corporation, Tokyo, Japan) according to the supplier’s instructions. Briefly, 50 μL of coloring solution was added to 50 μL of the collected media and incubated at 25 °C for 45 min. After dispensing 100 μL of 1 M HCl, the absorbance of the mixture was immediately measured at 570 nm using an iMark. Cytotoxicity was calculated using the following formula: Cytotoxicity (%) = (Abs in medium of cells treated with sudachitin or nobiletin)/(Abs in medium of cells treated with DMSO) × 100. LDH activity was calculated relative to the absorbance of DMSO-treated adipocytes.

### 2.5. Glycerol Release Assay

3T3-L1 adipocytes on day 8 were treated with 0.1% DMSO, 50 μM sudachitin, 50 μM nobiletin, or 10 μM isoproterenol (positive control) for 3 h. In pharmacological inhibition experiments, the cells were incubated for 1 h with 100 μM SQ22536 or 20 μM H-89 before treatment with sudachitin, nobiletin, or isoproterenol. At the end of each experiment, media were collected from each well to determine the concentration of free glycerol using the EnzyChrom^TM^ Glycerol Assay Kit (BioAssay Systems, Hayward, CA, USA) according to the supplier’s instructions. Briefly, 100 μL of working reagent containing glycerol kinase and glycerol phosphate oxidase was added to 10 μL of the collected media and incubated at 25 °C for 20 min. The absorbance of the reaction mixture was detected at 570 nm using an iMark.

### 2.6. Intracellular cAMP Levels Assay

3T3-L1 adipocytes on day 8 were treated with 0.1% DMSO, 30 μM sudachitin, 30 μM nobiletin, or 10 μM isoproterenol (positive control) for 15 min. After aspirating the culture medium, the adipocytes were treated with 0.1 M HCl and incubated at 25 °C for 20 min to completely lyse the adipocytes. Cell extracts were separated by centrifugation (MX-200, TOMY Seiko Co., Ltd., Tokyo, Japan) for 10 min at 1000× *g*, and the diluted supernatant was used in the assay. Intracellular cAMP levels were assessed using a Cyclic AMP ELISA Kit (Cayman Chemical, Ann Arbor, MI, USA) according to the supplier’s instructions.

### 2.7. Western Blotting

Protein levels were assessed by Western blotting as described previously [39]. Briefly, 3T3-L1 adipocytes were lysed in a radioimmunoprecipitation assay buffer containing protease (cOmplete, Roche Diagnostics, Indianapolis, IN, USA) and phosphatase inhibitor cocktails (PhosSTOP, Sigma-Aldrich Corp., St. Louis, MO, USA), and cell lysates (7 μg/lane) were subjected to 8 or 10% sodium dodecyl-sulfate polyacrylamide gel electrophoresis. The separated proteins were transferred to polyvinylidene difluoride membranes (BioRad). The membranes were incubated with 4% Block Ace (DS Pharma Biomedical Co., Ltd., Osaka, Japan) at 25 °C for 1 h, followed by treatment with primary antibodies (diluted 1:1000) at 4 °C for 16 h. The primary antibodies are listed in Table 1. Proteins were detected using ImmunoStar LD (FUJIFILM Wako Pure Chemical Industries, Tokyo, Japan) and a cooled charged-coupled camera system Light-capture II (ATTO Corp., Tokyo, Japan). For protein quantification, densitometric analysis of the blots was performed using ImageJ software (ver. 1.53f; https://imagej.nih.gov/ij/, accessed on 1 November 2022).

### 2.8. Statistical Analysis

All data are shown as means ± standard error of the mean (SEM) and were statistically analyzed using Excel-Toukei 2010 (Social Survey Research Information Co., Ltd., Osaka, Japan). One-way or two-way analysis of variance was followed by the Tukey–Kramer multiple-comparison test. *p* < 0.05 indicated a statistically significant difference.

## 3. Results

### 3.1. Cytotoxicity of Sudachitin and Nobiletin on 3T3-L1 Adipocytes

To determine the cytotoxicity of sudachitin and nobiletin on 3T3-L1 adipocytes, an LDH assay was performed, wherein the LDH released into the medium was quantified 24 and 48 h after treatment. There were no significant differences in LDH content in the medium among DMSO-, sudachitin-, and nobiletin-treated cells 24 and 48 h after treatment (Figure 1B,C). LDH content in the media containing 50 μM nobiletin-treated cells was 11% higher, but it was not significantly different from that of DMSO-treated cells after 24 h of treatment (Figure 1B). LDH content in the media containing cells treated with 10, 30, and 50 μM nobiletin was 15%, 19%, and 18% higher, respectively, than that of DMSO-treated cells after 48 h of treatment (Figure 1C). Sudachitin and nobiletin showed no cytotoxicity against 3T3-L1 adipocytes at 1–50 μM.

### 3.2. Sudachitin and Nobiletin Promoted Glycerol Release from 3T3-L1 Adipocytes

The levels of lipid storage were assessed in 3T3-L1 adipocytes treated with 50 μM sudachitin or nobiletin for 48 h. The lipid droplets in 3T3-L1 adipocytes decreased after treatment with sudachitin or nobiletin (Figure 2A). Lipid accumulation was 14% lower in adipocytes treated with nobiletin than in those treated with DMSO (Figure 2B). Sudachitin tended to reduce lipid accumulation when compared to cells treated with DMSO (*p* = 0.07). To elucidate the effects of sudachitin on lipolysis, the amount of glycerol in the 3T3-L1 adipocyte medium was assessed. Treatment with isoproterenol, as a positive control, resulted in a 6.2-fold increase in the medium concentrations of glycerol when compared to those treated with DMSO (Figure 2C). Treatment with nobiletin induced the release of glycerol from 3T3-L1 adipocytes, which is consistent with a previous report [36]. Sudachitin increased the concentration of glycerol in the medium by 2.0-fold when compared to that of cells treated with DMSO (Figure 2C). The levels of intracellular cAMP were also significantly upregulated by treatment with isoproterenol, sudachitin, and nobiletin when compared to treatment with DMSO (Figure 2D).

### 3.3. Time Course of Sudachitin- and Nobiletin-Induced Phosphorylation of PKA Substrates and HSL in 3T3-L1 Adipocytes

Next, the protein levels of phosphorylated PKA substrates and HSL at Ser563 and Ser660 were measured in 3T3-L1 adipocytes treated with 30 μM sudachitin or nobiletin for 0–120 min (Figure 3A,C). A significant increase in the protein levels of phosphorylated PKA substrates and phosphorylated HSL at Ser563 and Ser660 was detected within 5 min of sudachitin and nobiletin treatment (Figure 3B,D). Protein levels of PKA substrates phosphorylated and HSL phosphorylated at Ser660 were significantly higher in 3T3-L1 adipocytes treated with sudachitin and nobiletin for 5–120 min than those treated with DMSO. Treatment with sudachitin for 60 min resulted in a 1.6-fold increase in the protein levels of HSL phosphorylated at Ser563 when compared to those treated with DMSO (Figure 3B). Treatment with nobiletin for 5–120 min showed a 2.4-to-3.3-fold increase in protein levels of HSL phosphorylated at Ser563 when compared to those treated with DMSO for the corresponding time period (Figure 3D).

### 3.4. Induction of Phosphorylation of PKA Substrates and HSL by Treatment with Sudachitin and Nobiletin in a Dose-Dependent Manner

We investigated whether sudachitin or nobiletin dose-dependently induced the phosphorylation of PKA substrates and HSL at Ser563 and Ser660. Cells treated with sudachitin and nobiletin showed dose-dependent increases in the protein levels of PKA substrates phosphorylated and HSL phosphorylated at Ser563 and Ser660 (Figure 4A). In 3T3-L1 cells treated with 20, 30, and 50 μM sudachitin, protein levels of phosphorylated PKA substrates were 3.2-, 3.5-, and 4.8-fold higher, respectively, than those from cells treated with DMSO (Figure 4B). Treatment with 5, 10, 20, 30, and 50 μM nobiletin resulted in 2.4-, 3.2-, 4.4-, 4.8-, and 5.0-fold increases in the amount of phosphorylated PKA substrates, respectively. At 20 and 30 μM, the protein levels of phosphorylated PKA substrates were 39 and 38% higher, respectively, in adipocytes treated with nobiletin than in those treated with sudachitin. Protein levels of HSL phosphorylated at Ser563 increased 3.1- and 3.7-fold in cells treated with 30 and 50 μM sudachitin, respectively, when compared to those treated with DMSO (Figure 4B). Protein levels of HSL phosphorylated at Ser563 also increased 5.3-, 7.8-, and 8.5-fold in adipocytes treated with 20, 30, and 50 μM nobiletin, respectively, when compared to those treated with DMSO. At 20, 30, and 50 μM, protein levels of HSL phosphorylated at Ser563 were 123%, 154%, and 130% higher, respectively, in adipocytes treated with nobiletin than in those treated with sudachitin. Protein levels of HSL phosphorylated at Ser660 increased 2.6- and 3.4-fold in adipocytes treated with 30 and 50 μM sudachitin, respectively, when compared to those treated with DMSO (Figure 4B). Protein levels of HSL phosphorylated at Ser660 also increased by 2.6-, 4.6-, 5.9-, and 6.6-fold in adipocytes treated with 10, 20, 30, and 50 μM nobiletin, respectively, compared to those in adipocytes treated with DMSO. At 20, 30, and 50 μM, protein levels of HSL phosphorylated at Ser660 were 109%, 124%, and 97% higher, respectively, in adipocytes treated with nobiletin than in those treated with sudachitin.

### 3.5. Effects of AC and PKA Inhibition on Glycerol Release Induced by Sudachitin and Nobiletin

To better define the contribution of cAMP/PKA pathway activation to lipolysis induced by sudachitin and nobiletin, the effects of pharmacological inhibition of AC and PKA on glycerol release from adipocytes treated with sudachitin and nobiletin were investigated. Inhibition of AC and PKA did not affect the amount of glycerol released into the medium by DMSO-treated adipocytes (Figure 5). Glycerol release induced by sudachitin, nobiletin, and isoproterenol was significantly suppressed by pretreatment with the PKA inhibitor H-89 and the AC inhibitor SQ22536.

### 3.6. Effects of AC and PKA Inhibition on Phosphorylation of PKA Substrates and HSL Induced by Sudachitin and Nobiletin

The effects of pharmacological inhibition of AC and PKA on PKA substrate phosphorylation and HSL phosphorylation at Ser563 and Ser660 induced by sudachitin and nobiletin in 3T3-L1 adipocytes were also assessed (Figure 6A,C). Phosphorylation of PKA substrates and HSL at Ser563 and Ser660, stimulated by sudachitin and nobiletin, was reduced by pretreatment with SQ22536 (Figure 6B) and H-89 (Figure 6D). In addition, the effects of pharmacological inhibition of β3-AR, an upstream receptor of AC and PKA, on PKA substrate phosphorylation and HSL phosphorylation at Ser563 and Ser660 were determined. Pretreatment with L-748,337, a β3-AR-selective antagonist, had no effect on sudachitin- or nobiletin-induced phosphorylation of PKA substrates and HSL at Ser563 and Ser660, respectively (Appendix A).

## 4. Discussion

Intake of citrus peel polymethoxyflavones, such as sudachitin and nobiletin, prevents obesity in humans and rodents [25,26,27,40]. Nobiletin is known to promote lipolysis in adipocytes [14]. However, the effect of sudachitin on adipocyte lipolysis remains unclear. Herein, the effects of sudachitin on lipolysis and cAMP/PKA/HSL pathway activation in 3T3-L1 adipocytes are examined. Sudachitin and nobiletin induced glycerol release into the medium and increased the levels of intracellular cAMP, phosphorylated PKA substrates, and phosphorylated HSL. Pharmacological inhibition of AC and PKA prevented lipolysis and phosphorylation of PKA substrates and HSL stimulated by sudachitin and nobiletin. These findings support the notion that both nobiletin- and sudachitin-induced lipolysis are mediated by the activation of the cAMP/PKA/HSL pathway.

Sudachitin exerts anti-obesogenic effects in rodents and humans [25,26,27]. A previous study demonstrated that sudachitin suppresses HFD-induced obesity by stimulating mitochondrial biogenesis in myocytes. In a human study, supplementation with extracts of sudachi peel, including sudachitin, decreased the ratio of abdominal WAT to subcutaneous WAT when compared to supplementation with a placebo, thus resulting in a reduction in the risk of diabetes [25]. In humans, lipolysis and HSL activity are higher in visceral omental WAT than in subcutaneous WAT [41]. The current findings suggest that sudachitin decreased visceral fat by activating the cAMP/PKA/HSL pathway and adipocyte lipolysis.

Lipolysis is partly regulated by intracellular cAMP concentrations in response to lipolytic agents and hormones [42]. In the present study, sudachitin and nobiletin increased intracellular cAMP concentrations, which were inhibited by the AC inhibitor SQ22536 in 3T3-L1 adipocytes. These results indicate that the cAMP-dependent PKA pathway plays an important role in the lipolytic effect of sudachitin and nobiletin. However, the molecular mechanism of sudachitin- and nobiletin-induced increase in intracellular cAMP levels remains unclear. Pharmacological inhibition of β3-AR failed to reduce protein levels of phosphorylated PKA substrates and HSL in 3T3-L1 cells treated with sudachitin or nobiletin. It remains unknown whether HSL phosphorylation occurs through the activation of specific receptors or AC alone, thereby increasing cAMP levels and increasing PKA activity. Isoproterenol increases intracellular cAMP concentrations to promote lipolysis, whereas phosphodiesterase (PDE) activation hydrolyzes cAMP to inhibit PKA activity and lipolysis [42]. Nobiletin significantly reduces PDE activity at more than 100 μM in PC12D neuronal cells [43]. Insulin induces the activation of PDE3B in adipocytes in an Akt-dependent manner [44]. Herein, sudachitin reduced protein levels of the phosphorylated Akt in 3T3-L1 adipocytes [23,24]. Further experiments are required to elucidate the effects of sudachitin on PDE activity in adipocytes.

We demonstrated that sudachitin and nobiletin induced lipolysis through the cAMP/PKA/HSL pathway. ATGL is also a key lipase and regulates both basal and beta-adrenergic-stimulated lipolysis and catalyzes the first step in triglyceride hydrolysis, thereby resulting in diglyceride and fatty acid formation in adipocytes [45]. Perilipins are lipid droplet-related proteins that protect triglycerides from lipolysis [46]. PKA-induced phosphorylation of perilipin A at Ser517 is necessary for ATGL-dependent hydrolysis of triacylglycerol [47]. Perilipin binds to CGI-58, a co-regulator of ATGL activity [48], resulting in the suppression of ATGL activity. PKA-mediated phosphorylation of CGI-58 releases CGI-58 from perilipin [49]. These findings suggest that ATGL and HSL are involved in sudachitin- and nobiletin-induced lipolysis in 3T3-L1 cells.

Polymethoxyflavones exert various health effects. Nobiletin reduces inflammatory responses mediated by the induction of heme oxygenase-1, which is the primary antioxidant enzyme, expressed in 3T3-L1 and RAW264.7 cells [13]. Brite adipocytes are derived from white adipocytes and have an increased capacity for fatty acid oxidation [50]. 3T3-L1 adipocytes treated with 100 μM nobiletin show a brown fat-like phenotype, with an increased expression of *Ucp1* and fatty-acid-oxidation-associated genes [15]. Sinensetin (5,6,7,3′,4′-pentamethoxyflavone), a rare polymethoxyflavone found in certain citrus fruits, induces phosphorylation of PKA and HSL by increasing intracellular cAMP concentrations in 3T3-L1 adipocytes [51,52]. These results are consistent with sudachitin-induced lipolysis mediated by the activation of the cAMP/PKA/HSL pathway. The current findings demonstrate that nobiletin is a more potent inducer of the phosphorylation of PKA substrates and HSL than sudachitin. Sudachitin is structurally similar to nobiletin, but the three methoxy groups in nobiletin are replaced by hydroxy groups [18]. The previous study demonstrated that methylation of flavonoids increases in metabolic stability and cell membrane permeability [53]. The methoxy groups at the C-5, C-7, and C-4′ positions remarkably suppressed metabolic depletion in human liver homogenate and increased permeability in Caco-2 cells, the model of human intestinal absorption, when compared to the hydroxy groups at these positions [53]. These findings indicate that the methoxy groups at the C-5, C-7, and C-4′ positions are involved in the enhancement of the lipolytic action of nobiletin. 5-Hydroxy-3,6,7,8,3′,4′-hexamethoxyflavone is found exclusively in *Citrus sinensis* [6]. The methoxy group at the C-5 position in nobiletin was replaced with a hydroxy group, and the C-3 hydrogen was replaced with a methoxy group. 5-Hydroxy-3,6,7,8,3′,4′-hexamethoxyflavone increases intracellular cAMP levels and PKA activity in PC12 neuronal cells [54]. Thus, the number of methoxy groups and the methoxy groups in the C-7 and C-4′ positions may be associated with enhanced PKA activity induced by polymethoxyflavones in adipocytes.

In conclusion, these data demonstrate that sudachitin stimulates lipolysis in 3T3-L1 adipocytes. The induction of lipolysis by sudachitin and nobiletin was mainly mediated by the activation of the cAMP/PKA/HSL pathway. Notably, the lipolytic activity exerted by sudachitin may represent an effective mechanism for ameliorating obesity and related metabolic disorders.

## Figures and Tables

**Figure 1 foods-12-01947-f001:**
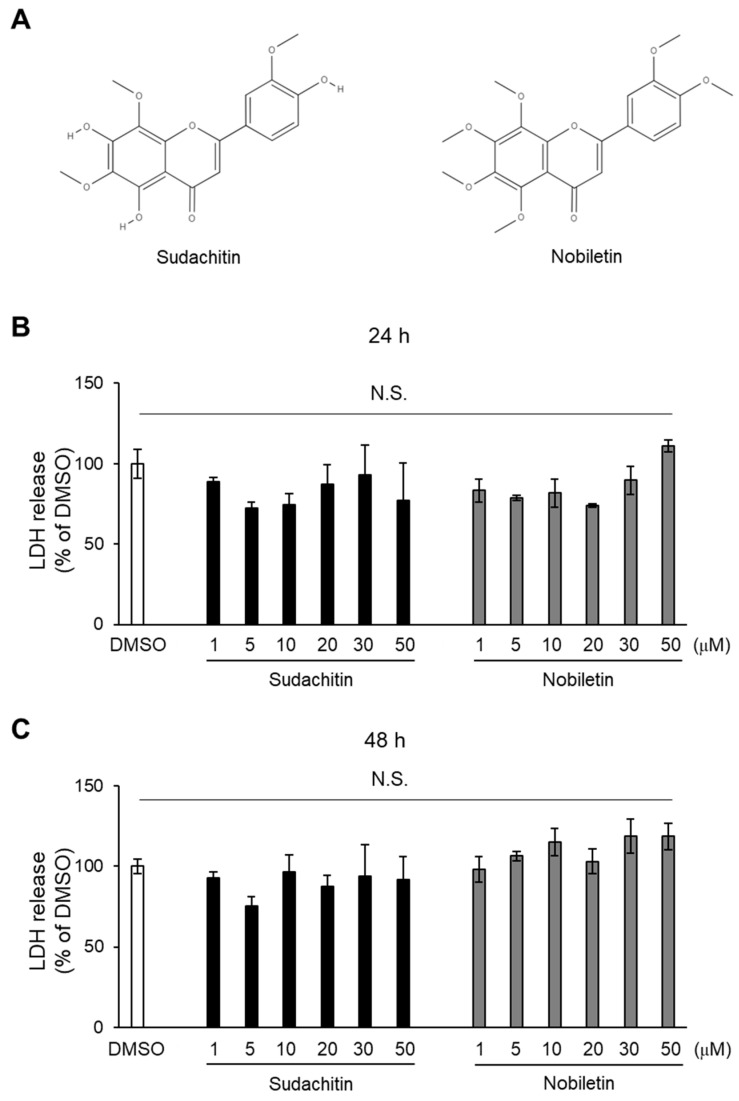
Cytotoxic effects of sudachitin and nobiletin. (**A**) Structures of sudachitin (5,7,4′-trihydroxy-6,8,3′-trimethoxyflavone) and nobiletin (5,6,7,8,3′,4′-hexamethoxyflavone). (**B**,**C**) 3T3-L1 adipocytes are treated with 0.1% DMSO, sudachitin, or nobiletin (1, 5, 10, 20, 30, and 50 µM) for 24 (**B**) and 48 h (**C**). Cytotoxicity is assessed using an LDH activity assay. Data are expressed as means ± SEM (*n* = 3). DMSO: dimethyl sulfoxide; LDH: lactate dehydrogenase; N.S.: not significant; SEM: standard error of the mean.

**Figure 2 foods-12-01947-f002:**
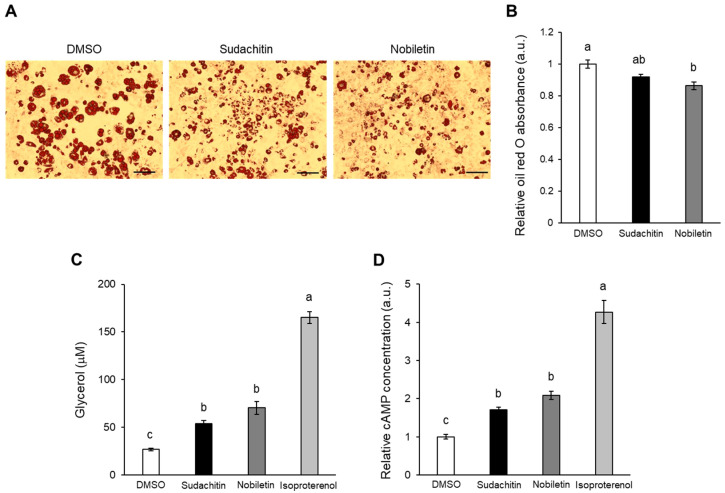
Induction of lipolysis and cAMP production by sudachitin or nobiletin treatment. Representative images (**A**) of and relative absorbance (**B**) of oil red O stained 3T3-L1 adipocytes. 3T3-L1 adipocytes treated with 0.1% DMSO and sudachitin or nobiletin (50 µM) for 48 h. Scale bar = 200 μm. (**C**) The glycerol content in the medium containing 3T3-L1 adipocytes treated with 0.1% DMSO, sudachitin, nobiletin (50 µM), or isoproterenol (10 µM) for 3 h. (**D**) cAMP concentrations measured in 3T3-L1 adipocytes treated with 0.1% DMSO, sudachitin, nobiletin (30 µM), or isoproterenol (10 µM) for 15 min. Data are expressed as fold induction relative to DMSO and are means ± SEM (*n* = 4). Different letters indicate significant differences (*p* < 0.05). a.u.: arbitrary unit; cAMP: cyclic AMP; DMSO: dimethyl sulfoxide; SEM: standard error of the mean.

**Figure 3 foods-12-01947-f003:**
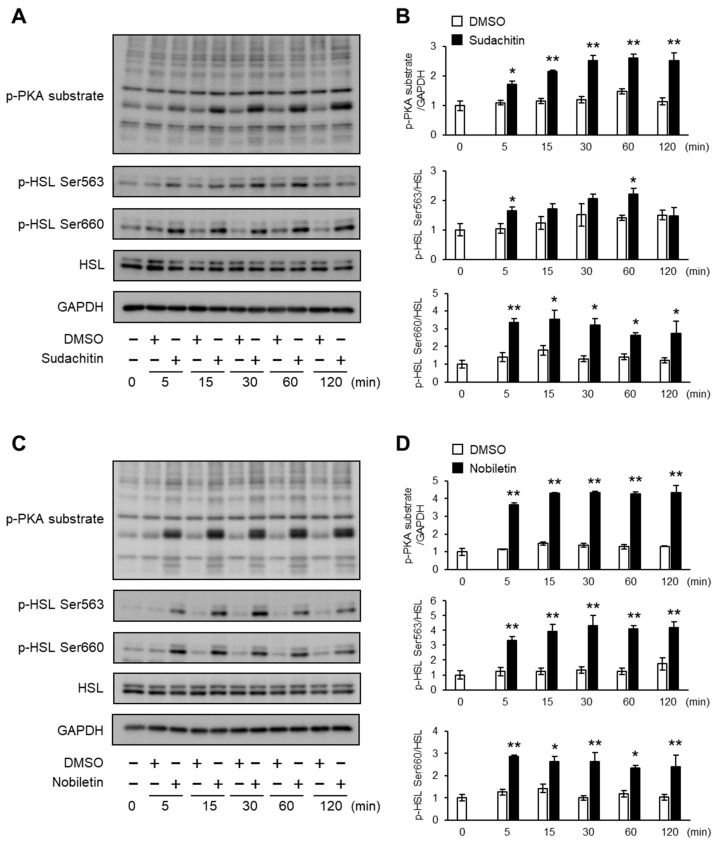
Time course of PKA/HSL pathway activation after treatment with sudachitin or nobiletin. (**A**,**C**) Protein levels of phosphorylated PKA substrates, HSL phosphorylated at Ser563, HSL phosphorylated at Ser660, total HSL, and GAPDH are measured in 3T3-L1 adipocytes at the indicated times after treatment with 0.1% DMSO, sudachitin (**A**), or nobiletin (**C**) (30 µM) for 0–120 min. (**B**,**D**) Densitometric ratios of phosphorylated PKA substrates/GAPDH and phosphorylated protein/total protein in 3T3-L1 adipocytes at the indicated times after treatment with 0.1% DMSO, sudachitin (**B**), or nobiletin (**D**) (30 µM). Data are expressed as means ± SEM (*n* = 4). * *p* < 0.05 and ** *p* < 0.01 versus DMSO at the corresponding time. DMSO: dimethyl sulfoxide; GAPDH: glyceraldehyde 3-phosphate dehydrogenase; HSL: hormone-sensitive lipase; PKA: protein kinase A; SEM: standard error of the mean.

**Figure 4 foods-12-01947-f004:**
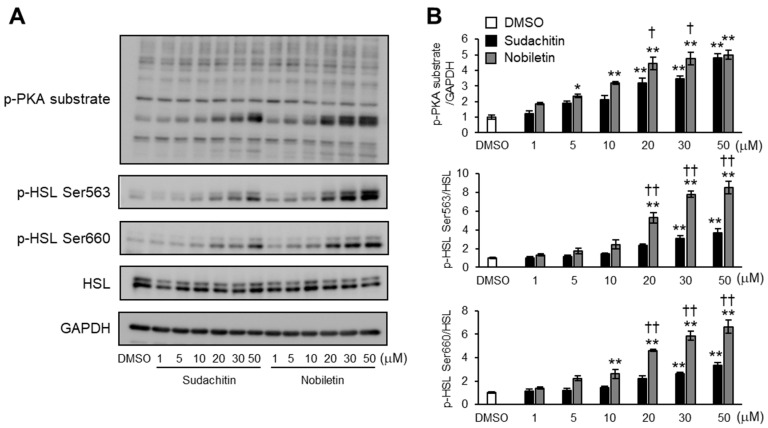
Does-dependent effects of sudachitin and nobiletin on PKA/HSL pathway activation. (**A**) Protein expression of phosphorylated PKA substrates, HSL phosphorylated at Ser563, HSL phosphorylated at Ser660, total HSL, and GAPDH measured in 3T3-L1 adipocytes treated with 0.1% DMSO, sudachitin, or nobiletin (0, 1, 5, 10, 20, 30, and 50 µM) for 1 h. (**B**) Densitometric ratios of phosphorylated PKA substrates/GAPDH and phosphorylated protein/total protein. Data are expressed as means ± SEM (*n* = 4). * *p* < 0.05 and ** *p* < 0.01 versus DMSO. † *p* < 0.05 and †† *p* < 0.01 versus sudachitin at the same concentration. DMSO: dimethyl sulfoxide; GAPDH: glyceraldehyde 3-phosphate dehydrogenase; HSL: hormone-sensitive lipase; PKA: protein kinase A; SEM: standard error of the mean.

**Figure 5 foods-12-01947-f005:**
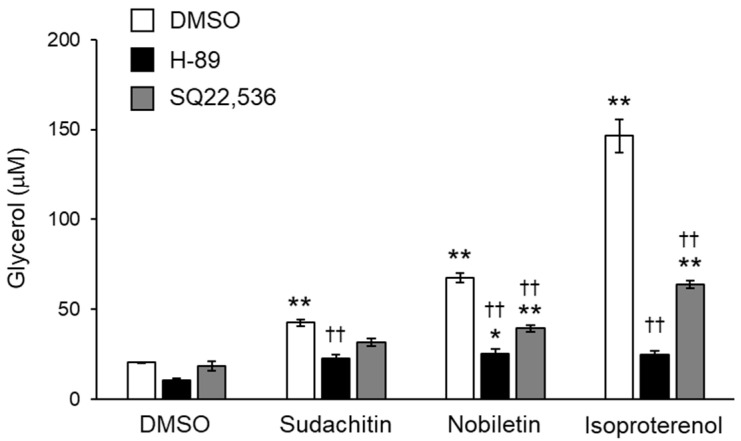
Inhibition of sudachitin- and nobiletin-induced lipolysis using AC and PKA inhibitors. Lipolysis is assessed by the glycerol content in the medium containing 3T3-L1 adipocytes treated with 0.05% DMSO, SQ22536 (100 µM), or H-89 (20 µM) for 1 h and then treated with 0.05% DMSO, sudachitin, nobiletin (50 µM), or isoproterenol (10 µM) for 3 h. Data are expressed as means ± SEM (*n* = 4). * *p* < 0.05 and ** *p* < 0.01 versus cells treated with DMSO and without inhibitors. †† *p* < 0.01 versus cells treated with the corresponding reagent and without inhibitors. AC: adenylate cyclase; DMSO: dimethyl sulfoxide; PKA: protein kinase A; SEM: standard error of the mean.

**Figure 6 foods-12-01947-f006:**
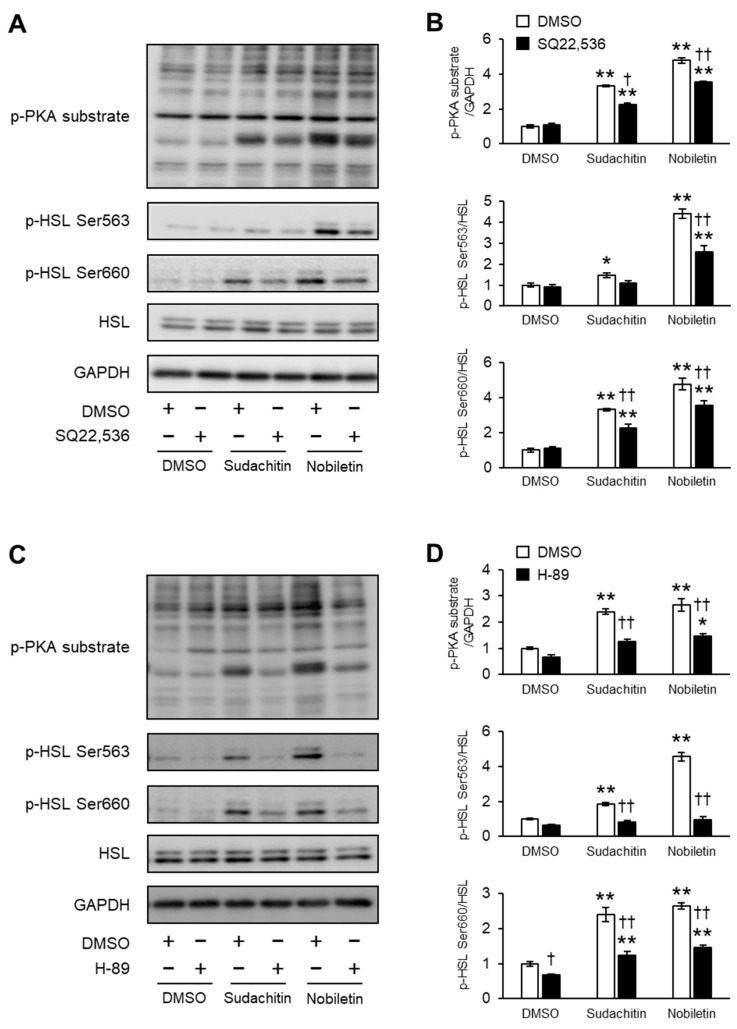
Inhibition of sudachitin- and nobiletin-induced phosphorylation of PKA substrates and HSL by AC and PKA inhibitors. (**A**,**C**) Protein levels of PKA substrates phosphorylated, HSL phosphorylated at Ser563, HSL phosphorylated at Ser660, total HSL, and GAPDH. 3T3-L1 adipocytes are treated with 0.05% DMSO, SQ22536 (100 µM) (**A**), or H-89 (20 µM) (**C**) before treatment with 0.05% DMSO, sudachitin, or nobiletin (30 µM). (**B**,**D**) Densitometric ratios of phosphorylated PKA substrates/GAPDH and phosphorylated protein/total protein. Data are expressed as means ± SEM (*n* = 4). * *p* < 0.05 and ** *p* < 0.01 versus cells treated with DMSO and without inhibitors. † *p* < 0.05 and †† *p* < 0.01 versus cells treated with the corresponding reagent and without inhibitors. DMSO: dimethyl sulfoxide; GAPDH: glyceraldehyde 3-phosphate dehydrogenase; HSL: hormone-sensitive lipase; PKA: protein kinase A; SEM: standard error of the mean.

**Table 1 foods-12-01947-t001:** Antibodies for Western blotting.

Antibodies	Source	Identifier
Anti-GAPDH	Novus Biologicals, LLC (Tokyo, Japan)	NB300-221
Anti-HSL	Cell Signaling Technology (Danvers, MA, USA)	#18381
Anti-phospho-HSL (Ser563)	Cell Signaling Technology	#4139
Anti-phospho-HSL (Ser660)	Cell Signaling Technology	#45804
Anti-phospho-PKA substrate	Cell Signaling Technology	#9624

GAPDH: glyceraldehyde-3-phosphate dehydrogenase; HSL: hormone-sensitive lipase; PKA: protein kinase A.

## Data Availability

All related data and methods are presented in this paper. Additional inquiries should be addressed to the corresponding author.

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
