# Peer review of "Sudachitin and Nobiletin Stimulate Lipolysis via Activation of the cAMP/PKA/HSL Pathway in 3T3-L1 Adipocytes"

_foods, 2023, doi:10.3390/foods12101947_

Round 1

Reviewer 1 Report

In this manuscript, authors determined the effects of sudachitin on lipolysis in 3T3-L1 adipocytes and found that sudachitin exerts its anti-obesogenic effects, at least in part, by inducing adipocyte lipolysis. 

After reviewing the manuscript, the following comments are derived:

Line 92. Briefly explain how this test (LDH-Cytotoxic Test Wako) was performed.

Line 93. Include the formula used and the absorbance the plate was read at. Might be worth mentioning results are expressed as percentage. 

Line 96. Please include if treatments were performed on day 8 or later (as done in line 80).

Line 102. As well as in line 92, briefly explain the usage of this kit.

Line 185: Figure 3 is missing in the manuscript. Only its footnote was included.

Line 330: Even though mechanisms and effects are properly explained, it would be interesting to include more information (if it is already available/published) about structure-mechanism relation for all the effects proven in this investigation/paper.

Quality of the research, figures, writing and overall content is satisfactory. The topic is relevant, current and interesting for general audience.

Author Response

We highly appreciate the valuable suggestions and comments from Reviewer 1. Please see the attachment.

Reviewer 2 Report

The manuscript "Sudachitin and nobiletin stimulate lipolysis via activation of the cAMP/PKA/HSL pathway in 3T3-L1 adipocytesn" highlight the effect of sudachitin and nobiletin in ameliorate obesity and insulin resistance. The manuscript is poorly written (in this form, the manuscript looks like an unfinished draft). Also, I recommend revising the English of the entire text. The information from the manuscript is rather confusing. Valuable knowledge must be added to the experimental design, results, and discussion chapter. Therefore, this paper needs drastic changes .

1.      The introduction chapter is very generally written. Please provide an adequate background for the work objectives. Consider highlighting more concretely the impact of sudachitin on obese individuals.

2.      Materials and Methods

-          What factors were considered in developing these protocols?

-          This section is quite confusing. Please rewrite to clarify. This section includes different materials: reagents, technical information, etc., without clear separation.

-          Please provide information about the equipment/apparatus used.

3.      Results chapter I recommend using the same unit of measure (µM, mL/mL) because it’s hard to follow the information.

4.      Line 138-146 I recommend checking the format of the text. Please check the journal guidelines.

5.      Figure 3 is missing.

6.      The discussion in this section is insufficient. There is no explanation of the obtained results and comparison with other authors. The authors should focus precisely on the results obtained and discuss them more deeply with specific original research from the domain.

Author Response

We highly appreciate the valuable suggestions and comments from Reviewer 2. Please see the attachment.

Reviewer 3 Report

Sudachitin and nobiletin, a kind of flavonoid, have been studied to reduce fat by mitochondrial biogenesis in skeletal muscle. It is also known to be effective in reducing ROS and inhibiting arthritis through activation of MAPK.

This thesis is evaluated as providing sufficient information to decompose fat cells.

However, I wonder why the Adipocyte size reduction assay was not conducted together with the study being conducted up to the Hormone-sensitive lipase (HSL) activity assay and evaluated as a good paper. In general, there are many reports that flavonoids affect HSL activity, so it is considered important to evaluate them. Please let me know if you have any results from using it.

I don't think there is a big problem with English quality.

Author Response

We highly appreciate the valuable suggestions and comments from Reviewer 3. Please see the attachment.

Round 2

Reviewer 2 Report

The paper was improved and all reviewers comments were addressed. 

Reviewer 3 Report

Requests for amendments and additions to this paper were faithfully reflected in the revised paper. Items to improve the quality of the thesis were included as revisions, which were faithfully supplemented. Therefore, the completeness of the thesis was much improved compared to the first paper, showing clear results.